# *word2ket*: Space-efficient Word Embeddings inspired by Quantum Entanglement

**Aliakbar Panahi**
Department of Computer Science
Virginia Commonwealth University
panahia@vcu.edu

**Seyran Saeedi**
Department of Computer Science
Virginia Commonwealth University
saeedis@vcu.edu

**Tom Arodz**[*]
Department of Computer Science
Virginia Commonwealth University
Richmond, VA 23284, USA
tarodz@vcu.edu

## Abstract

Deep learning natural language processing models often use vector word embeddings, such as word2vec or GloVe, to represent words. A discrete sequence of words can be much more easily integrated with downstream neural layers if it is represented as a sequence of continuous vectors. Also, semantic relationships between words, learned from a text corpus, can be encoded in the relative configurations of the embedding vectors. However, storing and accessing embedding vectors for all words in a dictionary requires large amount of space, and may strain systems with limited GPU memory. Here, we used approaches inspired by quantum computing to propose two related methods[1], *word2ket* and *word2ketXS*, for storing word embedding matrix during training and inference in a highly efficient way. Our approach achieves a hundred-fold or more reduction in the space required to store the embeddings with almost no relative drop in accuracy in practical natural language processing tasks.

## 1 Introduction

Modern deep learning approaches for natural language processing (NLP) often rely on vector representation of words to convert discrete space of human language into continuous space best suited for further processing through a neural network. For a language with vocabulary of size $d$, a simple way to achieve this mapping is to use one-hot representation – each word is mapped to its own row of a $d \times d$ identity matrix. There is no need to actually store the identity matrix in memory, it is trivial to reconstruct the row from the word identifier. Word embedding approaches such as word2vec (Mikolov et al., 2013) or GloVe (Pennington et al., 2014) use instead vectors of dimensionality $p$ much smaller than $d$ to represent words, but the vectors are not necessarily extremely sparse nor mutually orthogonal. This has two benefits: the embeddings can be trained on large text corpora to capture the semantic relationship between words, and the downstream neural network layers only need to be of width proportional to $p$, not $d$, to accept a word or a sentence. We do, however, need to explicitly store the $d \times p$ embedding matrix in GPU memory for efficient access during training and inference. Vocabulary sizes can reach $d = 10^5$ or $10^6$ (Pennington et al., 2014), and dimensionality of the embeddings used in current systems ranges from $p = 300$ (Mikolov et al., 2013; Pennington et al., 2014) to $p = 1024$ (Devlin et al., 2018). The $d \times p$ embedding matrix thus becomes a substantial, often dominating, part of the parameter space of a learning model.

In classical computing, information is stored in bits – a single bit represents an element from the set $\mathbb{B} = \{0, 1\}$, it can be in one of two possible states. A quantum equivalent of a bit, a qubit, is

---

[*]Corresponding author.
[1]PyTorch implementation available at: https://github.com/panaali/word2ket

fully described by a single two-dimensional complex unit-norm vector, that is, an element from the set $\mathbb{C}^2$. A state of an $n$-qubit quantum register corresponds to a vector in $\mathbb{C}^{2^n}$. To have exponential dimensionality of the state space, though, the qubits in the register have to be interconnected so that their states can become entangled; a set of all possible states of $n$ completely separated, independent qubits can be fully represented by $\mathbb{C}^{2n}$ instead of $\mathbb{C}^{2^n}$. Entanglement is a purely quantum phenomenon – we can make quantum bits interconnected, so that a state of a two-qubit system cannot be decomposed into states of individual qubits. We do not see entanglement in classical bits, which are always independent – we can describe a byte by separately listing the state of each of the eight bits. We can, however, approximate quantum register classically - store vectors of size $m$ using $O(\log m)$ space, at the cost of losing the ability to express all possible $m$-dimensional vectors that an actual $O(\log m)$-qubit quantum register would be able to represent. As we show in this paper, the loss of representation power does not have a significant impact on NLP machine learning algorithms that use the approximation approaches to store and manipulate the high-dimensional word embedding matrix.

## 1.1 Our Contribution

Here, we used approaches inspired by quantum computing to propose two related methods, *word2ket* and *word2ketXS*, for storing word embedding matrix during training and inference in a highly efficient way[2]. The first method operates independently on the embedding of each word, allowing for more efficient processing, while the second method operates jointly on all word embeddings, offering even higher efficiency in storing the embedding matrix, at the cost of more complex processing. Empirical evidence from three NLP tasks shows that the new *word2ket* embeddings offer high space saving rate at little cost in terms of accuracy of the downstream NLP model.

## 2 From Tensor Product Spaces to *word2ket* Embeddings

### 2.1 Tensor Product Space

Consider two separable[3] Hilbert spaces $\mathcal{V}$ and $\mathcal{W}$. A *tensor product space* of $\mathcal{V}$ and $\mathcal{W}$, denoted as $\mathcal{V} \otimes \mathcal{W}$, is a separable Hilbert space $\mathcal{H}$ constructed using ordered pairs $v \otimes w$, where $v \in \mathcal{V}$ and $w \in \mathcal{W}$. In the tensor product space, the addition and multiplication in $\mathcal{H}$ have the following properties

$$c\{v \otimes w\} = \{cv\} \otimes w = v \otimes \{cw\}, \tag{1}$$
$$v \otimes w + v' \otimes w = \{v + v'\} \otimes w,$$
$$v \otimes w + v \otimes w' = v \otimes \{w + w'\}.$$

The inner product between $v \otimes w$ and $v' \otimes w'$ is defined as a product of individual inner products

$$\langle v \otimes w, v' \otimes w' \rangle = \langle v, v' \rangle \langle w, w' \rangle. \tag{2}$$

It immediately follows that $||v \otimes w|| = ||v|| \, ||w||$; in particular, a tensor product of two unit-norm vectors, from $\mathcal{V}$ and $\mathcal{W}$, respectively, is a unit norm vector in $\mathcal{V} \otimes \mathcal{W}$. The Hilbert space $\mathcal{V} \otimes \mathcal{W}$ is a space of equivalence classes of pairs $v \otimes w$; for example $\{cv\} \otimes w$ and $v \otimes \{cw\}$ are equivalent ways to write the same vector. A vector in a tensor product space is often simply called a *tensor*.

Let $\{\psi_j\}$ and $\{\phi_k\}$ be orthonormal basis sets in $\mathcal{V}$ and $\mathcal{W}$, respectively. From eq. 1 and 2 we can see that

$$\left\{ \sum_j c_j \psi_j \right\} \otimes \left\{ \sum_k d_k \phi_k \right\} = \sum_j \sum_k c_j d_k \psi_j \otimes \phi_k,$$
$$\langle \psi_j \otimes \phi_k, \psi_{j'} \otimes \phi_{k'} \rangle = \delta_{j-j'} \delta_{k-k'},$$

where $\delta_z$ is the Kronecker delta, equal to one at $z = 0$ and to null elsewhere. That is, the set $\{\psi_j \otimes \phi_k\}_{jk}$ forms an orthonormal basis in $\mathcal{V} \otimes \mathcal{W}$, with coefficients indexed by pairs $jk$ and

---

[2]In Dirac notation popular in quantum mechanics and quantum computing, a vector $u \in \mathbb{C}^{2^n}$ is written as $|u\rangle$, and called a *ket*.

[3]That is, with countable orthonormal basis.

numerically equal to the products of the corresponding coefficients in $\mathcal{V}$ and $\mathcal{W}$. We can add any pairs of vectors in the new spaces by adding the coefficients. The dimensionality of $\mathcal{V} \otimes \mathcal{W}$ is the product of dimensionalities of $\mathcal{V}$ and $\mathcal{W}$.

We can create tensor product spaces by more than one application of tensor product, $\mathcal{H} = \mathcal{U} \otimes \mathcal{V} \otimes \mathcal{W}$, with arbitrary bracketing, since tensor product is associative. Tensor product space of the form

$$\bigotimes_{j=1}^{n} \mathcal{H}_j = \mathcal{H}_1 \otimes \mathcal{H}_2 \otimes \ldots \otimes \mathcal{H}_n$$

is said to have tensor *order*[4] of $n$.

## 2.2 Entangled Tensors

Consider $\mathcal{H} = \mathcal{V} \otimes \mathcal{W}$. We have seen the addition property $v \otimes w + v' \otimes w = \{v + v'\} \otimes w$ and similar property with linearity in the first argument – tensor product is bilinear. We have not, however, seen how to express $v \otimes w + v' \otimes w'$ as $\phi \otimes \psi$ for some $\phi \in \mathcal{V}$, $\psi \in \mathcal{W}$. In many cases, while the left side is a proper vector from the tensor product space, it is not possible to find such $\phi$ and $\psi$. The tensor product space contains not only vectors of the form $v \otimes w$, but also their linear combinations, some of which cannot be expressed as $\phi \otimes \psi$. For example, $\sum_{j=0}^{1} \sum_{k=1}^{1} \frac{\psi_j \otimes \phi_k}{\sqrt{4}}$ can be decomposed as $\left\{ \sum_{j=0}^{1} \frac{1}{\sqrt{2}} \psi_j \right\} \otimes \left\{ \sum_{k=1}^{1} \frac{1}{\sqrt{2}} \phi_k \right\}$. On the other hand, $\frac{\psi_0 \otimes \phi_0 + \psi_1 \otimes \phi_1}{\sqrt{2}}$ cannot; no matter what we choose as coefficients $a$, $b$, $c$, $d$, we have

$$\frac{1}{\sqrt{2}} \psi_0 \otimes \phi_0 + \frac{1}{\sqrt{2}} \psi_1 \otimes \phi_1 \neq (a\psi_0 + b\psi_1) \otimes (c\phi_0 + d\phi_1)$$
$$= ac\psi_0 \otimes \phi_0 + bd\psi_1 \otimes \phi_1 + ad\psi_0 \otimes \phi_1 + bc\psi_1 \otimes \phi_0,$$

since we require $ac = 1/\sqrt{2}$, that is, $a \neq 0$, $c \neq 0$, and similarly $bd = 1/\sqrt{2}$, that is, $b \neq 0$, $c \neq 0$, yet we also require $bd = ad = 0$, which is incompatible with $a, b, c, d \neq 0$.

For tensor product spaces of order $n$, that is, $\bigotimes_{j=1}^{n} \mathcal{H}_j$, tensors of the form $v = \bigotimes_{j=1}^{n} v_j$, where $v_j \in \mathcal{H}_j$, are called *simple*. Tensor *rank*[5] of a tensor $v$ is the smallest number of simple tensors that sum up to $v$; for example, $\frac{\psi_0 \otimes \phi_0 + \psi_1 \otimes \phi_1}{\sqrt{2}}$ is a tensor of rank 2. Tensors with rank greater than one are called *entangled*. Maximum rank of a tensor in a tensor product space of order higher than two is not known in general (Buczyński & Landsberg, 2013).

## 2.3 The *word2ket* Embeddings

A $p$-dimensional word embedding model involving a $d$-token vocabulary is[6] a mapping $f : [d] \to \mathbb{R}^p$, that is, it maps word identifiers into a $p$-dimensional real Hilbert space, an inner product space with the standard inner product $\langle \cdot, \cdot \rangle$ leading to the $L_2$ norm. Function $f$ is trained to capture semantic information from the language corpus it is trained on, for example, two words $i$, $j$ with $\langle f(i), f(j) \rangle \sim 0$ are expected to be semantically unrelated. In practical implementations, we represent $f$ as a collection of vectors $f_i \in \mathbb{R}^p$ indexed by $i$, typically in the form of $d \times p$ matrix $M$, with embeddings of individual words as rows.

We propose to represent an embedding $v \in \mathbb{R}^p$ of each a single word as an entangled tensor. Specifically, in *word2ket*, we use tensor of rank $r$ and order $n$ of the form

$$v = \sum_{k=1}^{r} \bigotimes_{j=1}^{n} v_{jk}, \tag{3}$$

---

[4] Note that some sources alternatively call $n$ a degree or a rank of a tensor. Here, we use tensor rank to refer to a property similar to matrix rank, see below.

[5] Note that some authors use rank to denote what we above called order. In the nomenclature used here, a vector space of $n \times m$ matrices is isomorphic to a tensor product space of order 2 and dimensionality $mn$, and individual tensors in that space can have rank of up to $\min(m, n)$.

[6] We write $[d] = \{0, ..., d\}$.

where $v_{jk} \in \mathbb{R}^q$. The resulting vector $v$ has dimension $p = q^n$, but takes $rnq = O\left(rq \log p/q\right)$ space. We use $q \geq 4$; it does not make sense to reduce it to $q = 2$ since a tensor product of two vectors in $\mathbb{R}^2$ takes the same space as a vector in $\mathbb{R}^4$, but not every vector in $\mathbb{R}^4$ can be expressed as a rank-one tensor in $\mathbb{R}^2 \otimes \mathbb{R}^2$.

If the downstream computation involving the word embedding vectors is limited to inner products of embedding vectors, there is no need to explicitly calculate the $q^n$-dimensional vectors. Indeed, we have (see eq. 2)

$$\langle v, w \rangle = \left\langle \sum_{k=1}^{r} \bigotimes_{j=1}^{n} v_{jk}, \sum_{k'=1}^{r} \bigotimes_{j=1}^{n} w_{jk'} \right\rangle = \sum_{k,k'=1}^{r,r} \prod_{j=1}^{n} \langle v_{jk}, w_{jk'} \rangle .$$

Thus, the calculation of inner product between two $p$-dimensional word embeddings, $v$ and $w$, represented via *word2ket* takes $O\left(r^2 q \log p/q\right)$ time and $O\left(1\right)$ additional space.

In most applications, a small number of embedding vectors do need to be made available for processing through subsequent neural network layers – for example, embeddings of all words in all sentences in a batch. For a batch consisting of $b$ words, the total space requirement is $O\left(bp + rq \log p/q\right)$, instead of $O\left(dp\right)$ in traditional word embeddings.

Reconstructing a $b$-word batch of $p$-dimensional word embedding vectors from tensors of rank $r$ and order $n$ takes $O\left(brpn\right)$ arithmetic operations. To facilitate parallel processing, we arrange the order-$n$ tensor product space into a balanced tensor product tree (see Figure 1), with the underlying vectors $v_{jk}$ as leaves, and $v$ as root. For example, for $n = 4$, instead of $v = \sum_k((v_{1k} \otimes v_{2k}) \otimes v_{3k}) \otimes v_{4k}$ we use $v = \sum_k(v_{1k} \otimes v_{2k}) \otimes (v_{3k} \otimes v_{4k})$. Instead of performing $n$ multiplications sequentially, we can perform them in parallel along branches of the tree, reducing the length of the sequential processing to $O\left(\log n\right)$.

Typically, word embeddings are trained using gradient descent. The proposed embedding representation involves only differentiable arithmetic operations, so gradients with respect to individual elements of vectors $v_{jk}$ can always be defined. With the balanced tree structure, *word2ket* representation can be seen as a sequence of $O\left(\log n\right)$ linear layers with linear activation functions, where $n$ is already small. Still, the gradient of the embedding vector $v$ with respect to an underlying tunable parameters $v_{lk}$ involves products $\partial\left(\sum_k \prod_{j=1}^{n} v_{jk}\right)/\partial v_{lk} = \prod_{j \neq l} v_{jk}$, leading to potentially high Lipschitz constant of the gradient, which may harm training. To alleviate this problem, at each node in the balanced tensor product tree we use LayerNorm (Ba et al., 2016).

## 3 LINEAR OPERATORS IN TENSOR PRODUCT SPACES AND *word2ketXS*

### 3.1 LINEAR OPERATORS IN TENSOR PRODUCT SPACES

Let $A : \mathcal{V} \to \mathcal{U}$ be a linear operator that maps vectors from Hilbert space $\mathcal{V}$ into vector in Hilbert space $\mathcal{U}$; that is, for $v, v', \in \mathcal{V}, \alpha, \beta \in \mathbb{R}$, the vector $A(\alpha v + \beta v') = \alpha Av + \beta Av'$ is a member of $\mathcal{U}$. Let us also define a linear operator $B : \mathcal{W} \to \mathcal{Y}$.

A mapping $A \otimes B$ is a linear operator that maps vectors from $\mathcal{V} \otimes \mathcal{W}$ into vectors in $\mathcal{U} \otimes \mathcal{Y}$. We define $A \otimes B : \mathcal{V} \otimes \mathcal{W} \to \mathcal{U} \otimes \mathcal{Y}$ through its action on simple vectors and through linearity

$$(A \otimes B) \left( \sum_{jk} \psi_j \otimes \phi_k \right) = \sum_{jk} (A\psi_j) \otimes (B\phi_k),$$

for $\psi_j \in \mathcal{V}$ and $\phi_k \in \mathcal{U}$. Same as for vectors, tensor product of linear operators is bilinear

$$\left( \sum_j a_j A_j \right) \otimes \left( \sum_k b_k B_k \right) = \sum_{jk} a_j b_k \left( A_j \otimes B_k \right).$$

In finite-dimensional case, for $n \times n'$ matrix representation of linear operator $A$ and $m \times m'$ matrix representing $B$, we can represent $A \otimes B$ as an $mn \times m'n'$ matrix composed of blocks $a_{jk}B$.

## 3.2 THE *word2ketXS* EMBEDDINGS

We can see a $p$-dimensional word embedding model involving a $d$-token vocabulary as a linear operator $F : \mathbb{R}^d \to \mathbb{R}^p$ that maps the one-hot vector corresponding to a word into the corresponding word embedding vector. Specifically, if $e_i$ is the $i$-th basis vector in $\mathbb{R}^d$ representing $i$-th word in the vocabulary, and $v_i$ is the embedding vector for that word in $\mathbb{R}^p$, then the word embedding linear operator is $F = \sum_{i=1}^d v_i e_i^T$. If we store the word embeddings a $d \times p$ matrix $M$, we can then interpret that matrix's transpose, $M^T$, as the matrix representation of the linear operator $F$.

Consider $q$ and $t$ such that $q^n = p$ and $t^n = d$, and a series of $n$ linear operators $F_j : \mathbb{R}^t \to \mathbb{R}^q$. A tensor product $\bigotimes_{j=1}^n F_j$ is a $\mathbb{R}^d \to \mathbb{R}^p$ linear operator. In *word2ketXS*, we represent the $d \times p$ word embedding matrix as

$$F = \sum_{k=1}^r \bigotimes_{j=1}^n F_{jk}, \tag{4}$$

where $F_{jk}$ can be represented by a $q \times t$ matrix. The resulting matrix $F$ has dimension $p \times d$, but takes $rnqt = O\left(rqt \max(\log p/q, \log d/t)\right)$ space. Intuitively, the additional space efficiency comes from applying tensor product-based exponential compression not only horizontally, individually to each row, but horizontally and vertically at the same time, to the whole embedding matrix.

We use the same balanced binary tree structure as in *word2ket*. To avoid reconstructing the full embedding matrix each time a small number of rows is needed for a multiplication by a weight matrix in the downstream layer of the neural NLP model, which would eliminate any space saving, we use lazy tensors (Gardner et al., 2018; Charlier et al., 2018). If $A$ is an $m \times n$ matrix and matrix $B$ is $p \times q$, then $ij^{th}$ entry of $A \otimes B$ is equal to

$$(A \otimes B)_{ij} = a_{\lfloor (i-1)/p \rfloor + 1, \lfloor (j-1)/q \rfloor + 1} b_{i - \lfloor (i-1)/p \rfloor p, j - \lfloor (j-1)/q \rfloor q}.$$

As we can see, reconstructing a row of the full embedding matrix involves only single rows of the underlying matrices, and can be done efficiently using lazy tensors.

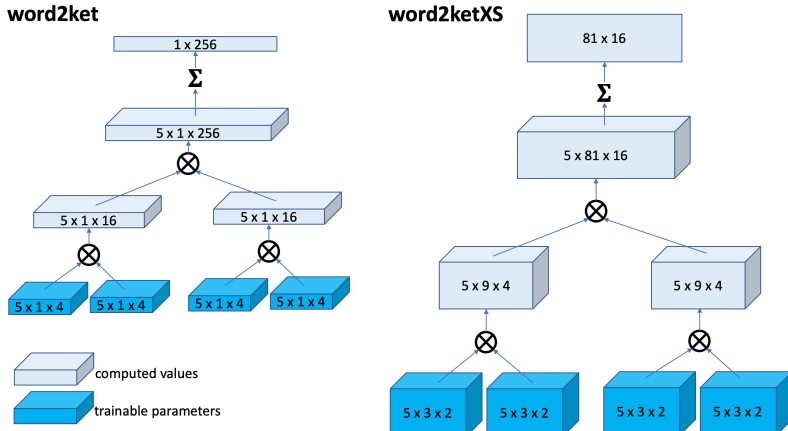

Figure 1: Architecture of the word2ket (left) and word2ketXS (right) embeddings. The word2ket example depicts a representation of a single-word 256-dimensional embedding vector using rank 5, order 4 tensor $\sum_{k=1}^5 \bigotimes_{j=1}^4 v_{jk}$ that uses twenty 4-dimensional vectors $v_{jk}$ as the underlying trainable parameters. The word2ketXS example depicts representation of a full 81-word, 16-dimensional embedding matrix as $\sum_{k=1}^5 \bigotimes_{j=1}^4 F_{jk}$ that uses twenty $3 \times 2$ matrices $F_{jk}$ as trainable parameters.

## 4 EXPERIMENTAL EVALUATION OF *word2ket* AND *word2ketXS* IN DOWNSTREAM NLP TASKS

In order to evaluate the ability of the proposed space-efficient word embeddings in capturing semantic information about words, we used them in three different downstream NLP tasks: text summarization, language translation, and question answering. In all three cases, we compared the accuracy in the downstream task for the proposed space-efficient embeddings with the accuracy achieved by regular embeddings, that is, embeddings that store $p$-dimensional vectors for $d$-word vocabulary using a single $d \times p$ matrix.

Table 1: Results for the GIGAWORD text summarization task using Rouge-1, Rouge-2, and Rouge-L metrics. The space saving rate is defined as the total number of parameters for the embedding divided by the total number of parameters in the corresponding regular embedding.

| Embedding | Order/Rank | Dim | RG-1 | RG-2 | RG-L | #Params | Space Saving Rate |
|---|---|---|---|---|---|---|---|
| Regular | 1/1 | 256 | 35.80 | 16.40 | 32.47 | 7,789,568 | 1 |
| word2ket | 4/1 | 256 | 33.65 | 14.87 | 30.47 | 486,848 | 16 |
| word2ketXS | 2/10 | 400 | 35.19 | 16.21 | 31.76 | 70,000 | 111 |
| word2ketXS | 4/1 | 256 | 34.05 | 15.39 | 30.75 | 224 | 34,775 |
| Regular | 1/1 | 8,000 | 36.71 | 17.48 | 33.37 | 243,424,000 | 1 |
| word2ketXS | 2/10 | 8000 | 35.17 | 16.35 | 31.72 | 19,200 | 12,678 |

In text summarization experiments, we used the GIGAWORD text summarization dataset (Graff et al., 2003) using the same preprocessing as (Chen et al., 2019), that is, using 200K examples in training. We used an encoder-decoder sequence-to-sequence architecture with bidirectional forward-backward RNN encoder and an attention-based RNN decoder (Luong et al., 2015), as implemented in PyTorch-Texar Hu et al. (2018). In both the encoder and the decoder we used internal layers with dimensionality of 256 and dropout rate of 0.2, and trained the models, starting from random weights and embeddings, for 20 epochs. We used the validation set to select the best model epoch, and reported results on a separate test set. We used Rouge 1, 2, and L scores (Lin, 2004). In addition to testing the regular dimensionality of 256, we also explored 400, and 8000, but kept the dimensionality of other layers constant.

The results in Table 1 show that word2ket can achieve 16-fold reduction in trainable parameters at the cost of a drop of Rouge scores by about 2 points. As expected, word2ketXS is much more space-efficient, matching the scores of word2ket while allowing for 34,000 fold reduction in trainable parameters. More importantly, it offers over 100-fold space reduction while reducing the Rouge scores by only about 0.5. Thus, in the evaluation on the remaining two NLP tasks we focused on word2ketXS.

The second task we explored is German-English machine translation, using the IWSLT2014 (DE-EN) dataset of TED and TEDx talks as preprocessed in (Ranzato et al., 2016). We used the same sequence-to-sequence model as in GIGAWORD summarization task above. We used BLEU score

Table 2: Results for the IWSLT2014 German-to-English machine translation task. The space saving rate is defined as the total number of parameters for the embedding divided by the total number of parameters in the corresponding regular embedding.

| Embedding | Order/Rank | Dimensionality | BLEU | #Params | Space Saving Rate |
|---|---|---|---|---|---|
| Regular | 1/1 | 256 | 26.44 | 8,194,816 | 1 |
| word2ketXS | 2/30 | 400 | 25.97 | 214,800 | 38 |
| word2ketXS | 2/10 | 400 | 25.33 | 71,600 | 114 |
| word2ketXS | 3/10 | 1000 | 25.02 | 9,600 | 853 |

Table 3: Results for the Stanford Question Answering task using DrQA model. The space saving rate is defined as the total number of parameters for the embedding divided by the total number of parameters in the corresponding regular embedding.

| Embedding | Order/Rank | F1 | #Params | Space Saving Rate |
|---|---|---|---|---|
| Regular | 1 | 72.73 | 35,596,500 | 1 |
| word2ketXS | 2/2 | 72.23 | 24,840 | 1,433 |
| word2ketXS | 4/1 | 70.65 | 380 | 93,675 |

to measure test set performance. We explored embedding dimensions of 100, 256, 400, 1000, and 8000 by using different values for the tensor order and the dimensions of the underlying matrices $F_{jk}$. The results in Table 2 show a drop of about 1 point on the BLEU scale for 100-fold reduction in the parameter space, with drops of 0.5 and 1.5 for lower and higher space saving rates, respectively.

The third task we used involves the Stanford Question Answering Dataset (SQuAD) dataset. We used the DrQA's model (Chen et al., 2017), a 3-layer bidirectional LSTMs with 128 hidden units for both paragraph and question encoding. We trained the model for 40 epochs, starting from random weights and embeddings, and reported the test set F1 score. DrQA uses an embedding with vocabulary size of 118,655 and embedding dimensionality of 300. As the embedding matrix is larger, we can increase the tensor order in word2ketXS to four, which allows for much higher space savings.

Results in Table 3 show a 0.5 point drop in F1 score with 1000-fold saving of the parameter space required to store the embeddings. For order-4 tensor word2ketXS, we see almost $10^5$-fold space saving rate, at the cost of a drop of F1 by less than two points, that is, by a relative drop of less than 3%. We also investigated the computational overhead introduced by the word2ketXS embeddings. For tensors order 2, the training time for 40 epochs increased from 5.8 for the model using regular embedding to 7.4 hours for the word2ketXS-based model. Using tensors of order 4, to gain additional space savings, increased the time to 9 hours. Each run was executed on a single NVIDIA Tesla V100 GPU card, on a 2 Intel Xeon Gold 6146 CPUs, 384 GB RAM machine. While the training time increased, as shown in Fig. 3, the dynamics of model training remains largely unchanged.

The results of the experiments show substantial decreases in the memory footprint of the word embedding part of the model, used in the input layers of the encoder and decoder of sequence-to-sequence models. These also have other parameters, including weight matrices in the intermediate layers, as well as the matrix of word probabilities prior to the last, softmax activation, that are not compressed by our method. During inference, embedding and other layers dominate the memory

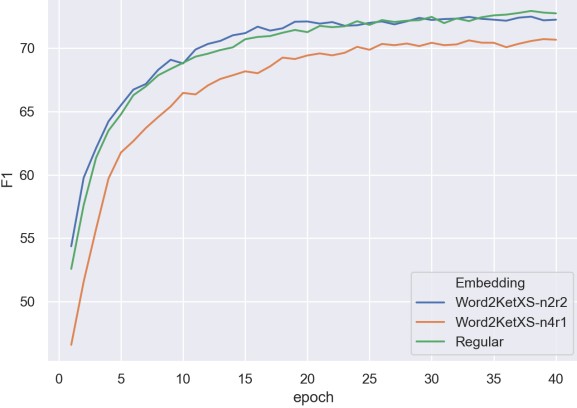

Figure 2: Dynamics of the test-set F1 score on SQuAD dataset using DrQA model with different embeddings: rank-2 order-2 word2ketXS, rank-1 order-4 word2ketXS, and regular embedding.

**CONTEXT:**
The 8- and 10-county definitions are not used for the greater Southern California Megaregion, one of the 11 megaregions of the United States. The megaregion's area is more expansive, extending east into Las Vegas, Nevada, and south across the Mexican border into Tijuana.

| Question | True Answers | Model Prediction |
|---|---|---|
| What is the name of the region that is not defined by the eight or 10 county definitions? | Southern California Megaregion, the greater Southern California Megaregion | greater Southern California Megaregion |
| How many megaregions are there in the United States? | 11 | 11 |
| What is the name of the state that the megaregion expands to in the east? | Nevada | Southern California Megaregion |
| Which border does the megaregion extend over? | Mexican | Tijuana |
| What is the name of the area past the border that the megaregion extends into? | Tijuana | Tijuana |

**CONTEXT:**
In 1900, the Los Angeles Times defined southern California as including "the seven counties of Los Angeles, San Bernardino, Orange, Riverside, San Diego, Ventura and Santa Barbara." In 1999, the Times added a newer county—Imperial—to that list.

| Question | True Answers | Model Prediction |
|---|---|---|
| Which newspaper defined southern California? | Los Angeles Times, the Los Angeles Times | Los Angeles Times |
| In which year did the newspaper define southern California? | 1900 | 1900 |
| In which year did the newspaper change its previous definition? | 1999 | 1900 |
| What was the newer county added to the list? | Imperial | Imperial |
| How many counties initially made up the definition of southern California? | seven | seven |

Figure 3: Test set questions and answers from DrQA model trained using rank-1 order-4 word2ketXS embedding that utilizes only 380 parameters (four $19 \times 5$ matrices $F_{jk}$, see eq. 4) to encode the full, 118,655-word embedding matrix.

footprint of the model. Recent successful transformer models like BERT by (Devlin et al., 2018), GPT-2 by (Radford et al., 2019), RoBERTa by (Liu et al., 2019) and Sparse Transformers by (Child et al., 2019) require hundreds of millions of parameters to work. In RoBERTa$_{\text{BASE}}$, 30% of the parameters belong to the word embeddings.

During training, there is an additional memory need to store activations in the forward phase in all layers, to make them available for calculating the gradients in the backwards phase. These often dominate the memory footprint during training, but one can decrease the memory required for storing them with e.g. gradient checkpointing Chen et al. (2016) used recently in Child et al. (2019).

## 4.1 RELATED WORK

Given the current hardware limitation for training and inference, it is crucial to be able to decrease the amount of memory these networks requires to work. A number of approaches have been used in lowering the space requirements for word embeddings. Dictionary learning (Shu & Nakayama, 2018) and word embedding clustering (Andrews, 2016) approaches have been proposed. Bit encoding has been also proposed Gupta et al. (2015). An optimized method for uniform quantization of floating point numbers in the embedding matrix has been proposed recently (May et al., 2019). To compress a model for low-memory inference, (Han et al., 2015) used pruning and quantization for lowering the number of parameters. For low-memory training sparsity (Mostafa & Wang, 2019) (Gale et al., 2019) (Sohoni et al., 2019) and low numerical precision (De Sa et al., 2018) (Micikevicius et al., 2017) approaches were proposed. In approximating matrices in general, Fourier-based approximation methods have also been used (Zhang et al., 2018; Avron et al., 2017). None of these approaches can mach space saving rates achieved by word2ketXS. The methods based on bit encoding, such as Andrews (2016); Gupta et al. (2015); May et al. (2019) are limited to space saving rate of at most 32 for 32-bit architectures. Other methods, for example based on parameter sharing Suzuki & Nagata (2016) or on PCA, can offer higher saving rates, but their storage requirement is limited by $d + p$, the vocabulary size and embedding dimensionality. In more distantly related work, tensor product spaces have been used in studying document embeddings, by using sketching of a tensor representing $n$-grams in the document Arora et al. (2018).

## ACKNOWLEDGMENTS

T.A. is funded by NSF grant IIS-1453658.

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
