# OpenReview forum: "word2ket: Space-efficient Word Embeddings inspired by Quantum Entanglement"
_ICLR.cc/2020/Conference — Accept (Spotlight)_

### Official Review · AnonReviewer1 · 2019-10-23
**Official Blind Review #1**

**Rating:** 8

**Review:**

The paper presents two methods to learn word embedding matrices that can be stored in much less space compared to traditional d x p embedding matrices, where d is the vocabulary size and p is the embedding size. Two methods are proposed: the first method estimates a p-dimensional embedding for a word as a sum of r tensor products of order n (tensor product of n q-dimensional embeddings).  This representation takes rnq parameters which can be much less than p, since p = q^n. The second method factorizes a full d x p embedding matrix jointly as a tensor product of much smaller t x q matrices and can obtain even larger space savings. Algorithms for efficiently computing full p-dimensional representations are also included. When only dot products are needed, the p-dimensional representations do not need to be explicitly constructed.

In my opinion the terminology from quantum computing and entanglement is an unnecessary complication. It would be better to simply talk about the special parametric form of the embeddings , which allows efficient storage. Tensor product representations have been used for embeddings before (but not with the goal of efficiency) (e.g. Arora et al 2018) https://openreview.net/pdf?id=B1e5ef-C-
The paper covers related work briefly and does not compare experimentally to any other work aiming to reduce memory usage for embedding models (e.g. using up-projection from lower-dimensional embeddings, or e.g. this paper: Learning Compact Neural Word Embeddings by Parameter Space Sharing by Suzuki and Nagata.

The experimental results on summarization, machine translation, and QA show that the methods can obtain comparable results to models using traditional word embeddings while obtaining savings of up to one-thousand fold decrease in space needed for the embeddings.

The experimental results seem to conflate the issues of the dimensionality of the word embeddings versus that of the higher layers. For example, in the summarization experiments, word2ketXS embeddings corresponding to 8000-dimensional embeddings are compared to a standard model with embeddings of size 256. The LSTM and layers for the word2ketXS model would become quite large but their size is not taken into account. In addition, the activation memory is often the major bottleneck and not the parameter memory. These issues are not discussed or made explicit in the experiments.

Overall the paper can be a strong contribution if the methods are stated with less quantum computing jargon, the overall parameter size and speed of the different models is specified in the experiments, and more specific connections to related work are made. Ideally, an experimental comparison to a prior method for space-efficient embeddings.

Question: What is the role of pre-trained Glove embeddings in the word2ket models? Was any pre-training done on unlabeled text?


Some typos:

Section 1.1

“matrix, as the cost ..”  -> “matrix, at the cost”

Under Eq (2)
I think you mean w instead of u

Section 3.2

F_j: R^t -> R^p , do you mean R^q


**Experience Assessment:**

I have published one or two papers in this area.

**Review Assessment: Checking Correctness Of Derivations And Theory:**

I assessed the sensibility of the derivations and theory.

**Review Assessment: Checking Correctness Of Experiments:**

I carefully checked the experiments.

**Review Assessment: Thoroughness In Paper Reading:**

I read the paper at least twice and used my best judgement in assessing the paper.

---

> ### Author Response · Authors · 2019-11-15
> **Response**
>
> Thank you for your detailed comments! We address the main points below:
>
> >> In my opinion the terminology from quantum computing and entanglement is an unnecessary complication. It would be better to simply talk about the special parametric form of the embeddings, which allows efficient storage.
>
> We removed some of the quantum computation connections from the introduction, keeping only enough to justify the title.
>
> >> Tensor product representations have been used for embeddings before (but not with the goal of efficiency) (e.g. Arora et al 2018) https://openreview.net/pdf?id=B1e5ef-C-
>
> We have expanded the “related work” section to include:
> In more distantly related work, tensor product spaces have been used in studying document embeddings, by using sketching of a tensor representing $n$-grams in the document \cite{arora2018compressed}.
>
> >> The paper covers related work briefly and does not compare experimentally to any other work aiming to reduce memory usage for embedding models (e.g. using up-projection from lower-dimensional embeddings, or e.g. this paper: Learning Compact Neural Word Embeddings by Parameter Space Sharing by Suzuki and Nagata.
>
> We have added a reference to Suzuki and Nagata’s (N&S) very interesting work. We note their experiments show substantial drop in quality on downstream tasks when the space-saving rate increases past 64. For a |U| x D embedding, their PS-SGNS method uses |U| B log K + C B K  F bits (N&S, section 3.3), where F is the number of bits (e.g. 32), C B and K are parameters, chosen to meet the assertion C B = D, and log K >=1 . Thus, their embeddings use |U| B log K + D K  F,  PS-SGNS method cannot use less then |U| + D memory. For the  DrQA / SQuAD experiments, we have |U|=118655, D=300, yet our method stores the embedding using just 380 floating numbers, a 380-fold reduction over the theoretical limit of PS-SGNS, with little impact on solution quality. Other existing methods, e.g. Uniform Quantization and K-means Compression  cannot offer more than 32 fold space reduction. K-means Compression “Compressing word embeddings” by Andrews et al. also has the same limit. “On the Downstream Performance of Compressed Word Embeddings” shows that PCA, which also has |U| + D memory, shows a big drop in performance after 4 fold compression. On the other hand, the minimum space requirement of our method is only 4 log |U|, if we use 2x2 matrix F_j and tensor of order n=log |U|. This logarithmic dependence on |U| translates to savings that grow higher with higher dictionary sizes.
>
> >> The experimental results seem to conflate the issues of the dimensionality of the word embeddings versus that of the higher layers.
>
> In all experiments, we kept the dimensions of the higher layers constant. The LSTM layers used the same 256 hidden size dimensions, but the embedding had a varying size. To clarify this, on page 6, we added "we also explored 400, and 8000 for the embedding size  but kept the dimensionality of other layers constant.”
>
> >> In addition, the activation memory is often the major bottleneck and not the parameter memory. These issues are not discussed or made explicit in the experiments.
>
> We have added two paragraphs following the experimental results to discuss the total memory breakdown during training and inference, and clarify where the savings are. During inference, there is no need for storing all the activations so embedding and other model’s parameters are the major bottlenecks. During training, one can decrease the memory required for storing activations with a method e.g. `gradient checkpointing` used recently in “Generating Long Sequences with Sparse Transformers” by Child et al., 2019.
>
> >>  Ideally, an experimental comparison to a prior method for space-efficient embeddings.
>
> We reduced the quantum part. We added classification to the description of the models:
> For the first two tasks, we now have “In both the encoder and the decoder we used internal layers with dimensionality of 256”. For the third task, we have “We used the DrQA's model, a 3-layer bidirectional LSTMs with 128 hidden units for both paragraph and question encoding.”. In terms of experimental comparison, as noted above, existing methods offer <64-fold embedding reduction, and the main goal of our method is to provide higher reduction rates.
>
> >> Question: What is the role of pre-trained Glove embeddings in the word2ket models? Was any pre-training done on unlabeled text?
>
> We did not use any pre-training of the models, and did not use pre-trained embeddings. To clarify this, we added “We trained the model for 40 epochs, starting from random weights and embeddings” to the experiments descriptions.
>
> >> Section 3.2 F_j: R^t -> R^p , do you mean R^q
> Indeed. Apologies for this and other typos.

---

### Official Review · AnonReviewer3 · 2019-10-24
**Official Blind Review #3**

**Rating:** 8

**Review:**


This paper proposes word2ket - a space-efficient form of storing word embeddings through tensor products. The idea is to factorize each d-dimensional vector into a tensor product of much smaller vectors (either with or without linear operators). While this results in a time cost for each word lookup, the space savings are enormous and can potentially impact several applications where the vocabulary size is too large to fit into processor memory (CPU or GPU). The experimental evaluation is done on several tasks like summarization, machine translation and question answering and convincingly demonstrates that one can achieve close to original model performance with very few parameters!

This approach would be very useful due to growing model sizes in many areas of NLP (e.g. large pre-trained models) and more broadly, deep learning.

Pros:
1. Novel idea, clear explanation of the method and the tensor factorization scheme.
2. Convincing experiments on a variety of NLP tasks that utilize word embeddings.

Cons:
1. (Minor) While this is not the focus of the paper, it would be useful to have at least one experiment with a state-of-the-art model on any of these tasks to further strengthen the results (most of the baseline models used currently seem to be below SOTA).


Minor comments:
Abstract: stain -> strain
Page 2: $||u|| \rightarrow ||w||$

**Experience Assessment:**

I do not know much about this area.

**Review Assessment: Checking Correctness Of Derivations And Theory:**

I did not assess the derivations or theory.

**Review Assessment: Checking Correctness Of Experiments:**

I assessed the sensibility of the experiments.

**Review Assessment: Thoroughness In Paper Reading:**

I read the paper at least twice and used my best judgement in assessing the paper.

---

> ### Author Response · Authors · 2019-11-15
> **Response**
>
> Thanks for your review and comments!
>
> >> Cons: 1. (Minor) While this is not the focus of the paper, it would be useful to have at least one experiment with a state-of-the-art model on any of these tasks to further strengthen the results (most of the baseline models used currently seem to be below SOTA).
>
> Indeed, these models have been surpassed by transformer-based models. We explored several models, and ultimately settled on training Bert.base, but it was not possible to advance far into the run given our computational budget. On V100 GPU at our disposal, it would take a month to train it. After two days of running it using regular embeddings and using word2ketXS 4/1 embedding, there training loss curves were indistinguishable. But at that stage, the optimization is still in its early stages, so this is not a conclusive finding.

---

### Official Review · AnonReviewer2 · 2019-10-28
**Official Blind Review #2**

**Rating:** 3

**Review:**


This paper explores two related methods to reduce the number of parameters required (and hence the memory footprint) of neural NLP models that would otherwise use a large word embedding matrix. Their method, inspired by quantum entanglement, involves computing word embeddings on-the-fly (or by directly computing the output of the "word embedding" with the first linear layer of network). They demonstrate their method can save an impressive amount of memory and does not exhibit big performance losses on three nlp tasks that they explore.

This paper is clearly written (with only a couple of typos) but does not yet reach publication standard. Whilst the empirical performance of their approach is promising from the perspective of saving reducing memory requirements, more experiments are required and more careful comparisons to baselines and other methods in the literature for saving memory/parameters. In general the related work and experimental sections are weak and brief, with only superficial analysis. There is  lack of careful analysis and insight into their results, as well as a careful comparisons to other work in this area.

The choice of tasks to evaluate on is broad, which is a strength, but is missing simpler tasks that one would expect to see, such as a text classification dataset, or simple bag-of-vectors style models. In addition, the choice of models are somewhat outdated baselines. It seems that transformers would be an ideal setting for their approach, as transformers have rather high dimensional word embedding matrices, but the authors do not run experiments with the now-ubiquitous Transformer.

The quantum inspiration is largely a distraction, and I think the paper would benefit from this element being scaled back or removed in order to free up space for more experiments.

The authors acknowledge one key weakness of their approach, that both training and inference time are increased (by 28% or 55% longer for DocQA depending on compression)  but much more work could be done to understand the best way to  mitigate for longer training and inference times.

The authors argue that reducing the memory footprint of models is vital to address hardware limitations for training and inference for large models like BERT or ROBERTA, but this argument is not particularly strong. Generally current limitations for training these kinds of models  are the long training times and being able to fit large batches onto our hardware, and the vocabulary matrix is only a constant factor here. And since training time is a bottleneck, the added value of saving memory vs slowing the training speed by 30-50% is debatable.

Here are some questions for the authors that come to mind when reviewing:

How does your method compare to other published methods on your benchmarks?

which choices for r and k lead to the best time/memory/performance tradeoff? how does this compare to other compression methods (on your tasks)

Seq2Seq models usually involve multiplying the the output hidden state with a vocab matrix before softmaxing over all the vocabulary produce word probabilities - did you account for this? Does your method work for the output vocab matrix?

Did you investigate pre-training word2ket like word2vec or Glove?


**Experience Assessment:**

I do not know much about this area.

**Review Assessment: Checking Correctness Of Derivations And Theory:**

I assessed the sensibility of the derivations and theory.

**Review Assessment: Checking Correctness Of Experiments:**

I carefully checked the experiments.

**Review Assessment: Thoroughness In Paper Reading:**

I read the paper thoroughly.

---

> ### Author Response · Authors · 2019-11-15
> **Response**
>
> Thank you for your detailed comments! We address the main points below:
>
> >> The choice of tasks to evaluate on is broad, which is a strength, but is missing simpler tasks that one would expect to see, such as a text classification dataset, or simple bag-of-vectors style models.
>
> Following the suggestion, we trained GloVe using regular embedding and word2ketXS 4/1 for 600K steps on enwiki8 dataset. The evaluation loss started at 0.75 and flattened to 0.03 for word2ketXS and 0.01 for regular embedding.
>
> >> authors do not run experiments with the now-ubiquitous Transformer.
>
> Training transformers from scratch is not possible with our current computational budget as it takes more than a month to train using a single V100 GPU. We ran an experiment with Bert.base for 2 days using regular and word2ketXS 4/1 embedding. The difference at this stage is infinitesimal.
>
> >> much more work could be done to understand the best way to mitigate for longer training and inference times.
>
> From our experiments with pre-training Bert, which is larger model than the ones we used in our experiments, the increase in time dropped to 7%.
>
> >> Generally current limitations for training these kinds of models  are the long training times and being able to fit large batches onto our hardware, and the vocabulary matrix is only a constant factor here.
>
> We have added two paragraphs following the experimental results to discuss the total memory breakdown during training and inference, and clarify where the savings are. During inference, there is no need for storing all the activations so embedding and other model’s parameters are the major bottlenecks. During training, one can decrease the memory required for storing activations with a method e.g. `gradient checkpointing` proposed in “Training deep nets with sublinear memory cost.” by Chen et. al. 2016 and used recently in “Generating Long Sequences with Sparse Transformers” by Child et al., 2019. Other approaches, such as “ALBERT: A Lite BERT for Self-supervised Learning of Language Representations” by Lan et. al. use matrix factorization that gives 5 to 30 fold reduction for the base and xxlarge model.  “Low-Memory Neural Network Training: A Technical Report” by Sohoni reports 8 to 60 fold reduction in the peak memory required to train a model for a DynamicConv Transformer and WideResNet model by combining methods such as (1) imposing sparsity on the model, (2) using low precision, (3) microbatching, and (4) gradient checkpointing.
>
> >> Here are some questions for the authors that come to mind when reviewing:
>
> >> How does your method compare to other published methods on your benchmarks?
>
> The obstacle to comparing published methods for word embedding compression empirically with outs is that existing methods have hard limits on the compression rate. E.g. bit-reductions techniques can only reduce 32bits to 1bit. Other methods also have hard limits on their storage requirement, for example PS-SGNS method cannot use less then |U| + D memory. For the  DrQA / SQuAD experiments, we have |U|=118655, D=300, yet our method stores the embedding using just 380 floating numbers, a 380-fold reduction over the theoretical limit of PS-SGNS, with little impact on solution quality. We have expanded Related Work section to comment on this issue.
>
> >> which choices for r and k lead to the best time/memory/performance tradeoff? how does this compare to other compression methods (on your tasks)
>
> The compression rate depends on the tensor product rank and order. Increasing the order leads to logarithmic compression, while increasing the rank reduces the compression by a linear rate. Increasing order and reducing rank both lead to lower flexibility in what the compressed model can approximate. In the Giga experiment, we reported embedding dim of 400 with order of 2 and rank 10. We also investigated ranks ranging from 1 to 128. Reducing rank below 8 leads to observable drop in accuracy, of about e.g. RG-1 drops from 35.17 to about 34. Increasing the rank past 10 does not increase accuracy.
>
> >> Seq2Seq models usually involve multiplying the the output hidden state with a vocab matrix before softmaxing over all the vocabulary produce word probabilities - did you account for this?  Does your method work for the output vocab matrix?
>
>
> No. Neither our method nor other methods aim at compressing this matrix. We added two paragraphs at the end of experimental results section to clarify this and other memory considerations for training and inference. In Transformer, this matrix is shared with the embedding matrix, in principle we can use the same lazy tensor approach to utilize the transposed embeddings matrix without explicitly reconstructing it.
>
>
> >> Did you investigate pre-training word2ket like word2vec or Glove?
> No, we trained all models from random initializations. We added a clarification highlighting that to the manuscript.

---

### Decision · Program_Chairs · 2019-12-19

**Decision:**

Accept (Spotlight)

**Comment:**

This paper proposes quantum-inspired methods for increasing the parametric efficiency of word embeddings. While a little heavy in terms of quantum jargon, and perhaps a little ignorant of loosely related work in this sub-field (e.g. see the work of Coecke and colleagues from 2008 onwards), the majority of reviewers were broadly convinced the work and results were of sufficient merit to be published.